# Magnesium Nutritional Status, Risk Factors, and the Associations with Glucose Parameters of Childbearing Women in the China Adult Chronic Disease and Nutrition Surveillance (2015)

**DOI:** 10.3390/nu14040847

**Published:** 2022-02-17

**Authors:** Huidi Zhang, Yang Cao, Qingqing Man, Yuqian Li, Shanshan Jia, Rui Wang, Jiaxi Lu, Lichen Yang

**Affiliations:** National Institute for Nutrition and Health, Chinese Center for Disease Control and Prevention, Beijing 100050, China; zhanghuidi1114@126.com (H.Z.); yasmine0814@163.com (Y.C.); manqq@ninh.chinacdc.cn (Q.M.); liyq@ninh.chinacdc.cn (Y.L.); jiass@ninh.chinacdc.cn (S.J.); wangr@ninh.chinacdc.cn (R.W.); lujx@ninh.chinacdc.cn (J.L.)

**Keywords:** magnesium status, magnesium deficiency, childbearing women

## Abstract

Magnesium is an essential element and participates in many metabolic pathways. Inadequate magnesium levels may lead to various health problems such as type 2 diabetes (T2DM), hypertension, and cancer. But the role of Mg in childbearing women of China is still a relatively narrow researched field. We aimed to assess the Mg nutritional status, explore the risk factors of Mg deficiency, and the associations between Mg and glucose parameters among childbearing women in a nationally representative sample. A total of 1895 18–44 years childbearing women were recruited from the China Adult Chronic Disease and Nutrition Surveillance (2015). Multivariate logistic regression was used to explore the risk factors for Mg deficiency and estimate the odds ratios (ORs) and 95% confidence intervals (95% CIs) for the risk of hyperglycemia. The mean value of Mg was 0.87 mmol/L and the prevalence of deficiency was 4.69%. The risk factors of Mg deficiency (Mg < 0.75 mmol/L) was city-type of rural (*p* = 0.045), while calcium (*p* = 0.001), LDL-C (*p* = 0.024), age group of 26–35 years (*p* = 0.016), 36–44 years (*p* = 0.006), and CNNM2 rs3740393 genotypes of GC (*p* = 0.027) were protective factors. It was also found that magnesium deficiency induces an increase in plasma glucose (*p* = 0.001). Compared with the reference range, Mg < 0.75 mmol/L would have a 6.53 fold risk for T2DM, a 5.31 fold risk for glucose-hyperglycemia, and a 9.60 fold risk for HbA1c-hyperglycemia. Consistently, there was a negative association between plasma Mg and blood glucose parameters in the dose–response study. More attention should be paid to the nutritional status of magnesium and the impact of magnesium deficiency on human health.

## 1. Introduction

Magnesium is an essential element required as a co-factor for over 600 enzymatic reactions [1,2]. It is required for the biochemical functions of many metabolic pathways, such as the production of adenosine triphosphate (ATP), the regulation of calcium and potassium transport across membranes, and the stabilization of secondary structures of DNA and RNA [3,4].

A moderate deficiency of Mg can cause signs of weakness, loss of appetite, nausea, and vomiting [5]. When magnesium deficiency is more severe, it can lead to disturbances of the muscular, gastrointestinal, cardiovascular, and metabolic systems [6], and the signs of severe deficiency also could be indicated for chronic latent magnesium deficiency. According to the WHO report, subclinical defects of magnesium are common in developed and developing countries [7]. It is reported that the prevalence of magnesium deficiency ranges from 2.5 to 15% when 0.75 mmol/L is used as the cut-off value [8]. In terms of gender differences, the incidence rate of female hypomagnesemia is higher than that of men [9]. Research suggests that magnesium deficiency increases the risk of type 2 diabetes, which is a major public health problem in the world [10].

To date, no nationally representative data on magnesium nutrition status among women of reproductive age in China have been reported. The aim of this study is to assess magnesium nutritional status and the risk factors of magnesium deficiency in Chinese childbearing women. At the same time, we will preliminarily explore the relationship between magnesium and glucose-related parameters.

## 2. Materials and Methods

### 2.1. Subjects

The study was based on the 2015 Chinese Adult Chronic Disease and Nutrition Surveillance, which was a nationally representative cross-sectional survey. All participants in the survey were selected using a stratified, multi-stage, and probabilistic random sampling scheme. The sampling method was described in detail in Yu et al. [11]. In this study, subjects of reproductive age were randomly selected from the entire population, taking into account regional types and monitoring sites. We excluded samples with incomplete questionnaire information, poor blood quality (such as hemolysis), and biological index values below the detection limit. In total, 1895 childbearing women were finally included in this study. Written informed consent was obtained from every participant. The study was conducted in accordance with the Declaration of Helsinki and the protocol was approved by the Ethics Committee of the National Institute of Nutrition and Health, Chinese Center for Disease Control and Prevention (file number 201519-A).

### 2.2. Basic Information and Sample Collection

Basic demographic information was collected through a standardized questionnaire and filled by uniformly trained investigators. Based on a self-reported questionnaire—nationality, education, and drinking situation were recorded. The education level was divided into primary (primary school and below), medium (junior high school/high school/secondary school), and advanced (junior college or above). The district was divided into eastern, central, and western. The anthropometric measurements were also performed by trained medical staff following standardized procedures. The physical examination consists of weight, height, waist circumference, and blood pressure. Body mass index (BMI) was calculated by weight and height.

Venous blood was collected in the morning after at least 10 h of fasting, and each blood sample was divided into an anticoagulation tube and plasma separator tube. Immediately after blood collection, the blood sample in the plasma separation tube was centrifuged at 3000× *g* for 15 min, divided into aliquots, and frozen at −80 °C for subsequent analysis.

### 2.3. Plasma Mg and Laboratory Index Detection and Evaluation Standards

Serum fasting glucose, high-density lipoprotein cholesterol (HDL-C), low-density lipoprotein cholesterol (LDL-C), total cholesterol (TC), and triglyceride (TG) were measured enzymically using an automatic biochemical analyzer (Hitachi 7600, Tokyo, Japan). Glycated hemoglobin (HbA1c) was determined by high-performance liquid chromatography (HPLC) using Trinity Biotech, Premier Hb9210 (Dublin, Ireland).

The definitions of type 2 diabetes, glucose-hyperglycemia, and HbA1c-hyperglycemia met the respective diagnostic criteria recommended by the World Health Organization in 2006 [12]. Type 2 diabetes was defined as impaired fasting glucose (FPG ≥ 7.0 mmol/L) and/or 2-h post-glucose load ≥11.1 mmol/L and/or HbA1c ≥ 6.5%. Glucose-hyperglycemia was defined as impaired glucose tolerance (FPG ≥ 6.1 mmol/L and < 7.0 mmol/L). HbA1c-hyperglycemia was defined as HbA1c ≥ 6.5%.

Mg and Ca concentrations were measured in our laboratory by inductively coupled plasma mass spectrometry (ICP-MS, PerkinElmer, NexION 350, Waltham, MA, USA). Plasma element was measured by 0.5% (*v*/*v*) high-purity nitric acid dilution (1:20). The precision and accuracy of the analysis were monitored at 10-sample intervals using the quality control samples (Seronorm, Level-2, Billingstad, Norway).

The coefficient of variation between and within batches of Mg was 2.33 and 1.19%, respectively. The coefficient of variation of Ca between and within batches was 1.23 and 2.62%, respectively. The recovery of Mg and Ca were 100.10 and 97.63%, respectively. According to the current cut-offs of the plasma magnesium concentration for assessing magnesium status [13], the concentration of Mg (mmol/L) was divided into three parts: ≤0.75 as hypomagnesemia, 0.75–0.85 as chronic latent magnesium deficiency, ≥0.85 as magnesium sufficient.

### 2.4. SNP Selection and Detection

We selected genes according to the following criteria: (1) biological importance in magnesium metabolism, transport or degradation; (2) genome-wide association studies (GWAS) indicates genetic variation significantly associated with magnesium, or genetic variation that has been applied to Mendelian randomization studies and magnesium supplementation intervention tests. The selected base is CNNM2. For the studied genes, single nucleotide polymorphisms (SNPs) were labeled from the International HapMap project (http://www.hapmap.org/cgi-perl/gbrowse/hapmap3_B36, accessed on 15 September 2021). The SNP was selected to meet the requirements of heterozygosity (secondary allele frequency [MAFS] > 0.1). The selected SNP was rs3740393. SNP genotyping was detected by the improved multiple ligase detection reaction (iMLDR).

### 2.5. Data Analyses

Statistical analyses were performed using SPSS version 19.0 and SAS 9.3. The difference of Mg status among different groups was analyzed using the Kruskal–Wallis test or one-way ANOVA for continuous variables and Chi-square test for categorical variables. Frequencies were presented as percentages (%), and the prevalence rates of subgroups were compared by the Chi-square test. Multivariate logistic regression was used to explore the risk factors of Mg deficiency. The multinomial logistic regression model was also used to assess the associations of glucose parameters with plasma magnesium concentrations, which could be described as the odds ratios (ORs) and 95% confidence intervals (95% CIs). All statistical tests were two-sided and statistical significance was considered at *p* < 0.05.

## 3. Results

### 3.1. Plasma Magnesium Concentrations of 1895 Childbearing Women

This study was conducted on 1895 childbearing women after excluding the hemolysis, incomplete data, and other unqualified samples. The age of this population was 31.04 ± 7.61 years. The plasma Mg concentration distributions in different subgroups are shown in Table 1. There was a significant difference in Mg concentrations between district subgroups, and the difference could also be found with genotypes of rs3740393 in the CNNM2 gene. However, no such difference was found in other characteristics such as age group, city-type, nationality, BMI, education, and drink.

### 3.2. Comparison of Mg Status among Different Groups in 1895 Subjects

The comparison of Mg status among different groups is shown in Table 2. The overall prevalence of plasma Mg deficiency, insufficiency, and sufficiency was 4.69, 33.09, and 62.22%, respectively. The status of Mg was marginal different in city-type subgroups (*p* = 0.054) and CNNM2 rs3740393 genotypes (*p* = 0.063), and there was no difference in the status of magnesium in other stratified subgroups.

### 3.3. Multivariate Logistic Regression Model for Risk Factors Associated with Mg Deficiency

The exploration of the risk factors associated with Mg deficiency is shown in Table 3. Among the continuous variables, calcium and LDL-C were found to be protective factors for magnesium deficiency; the odds ratio (OR) value was 0.01 (*p* = 0.001) and 0.35 (*p* = 0.024), respectively. However, magnesium deficiency was found to induce an increase in plasma magnesium; the OR value was 1.61 (*p* = 0.001). Among the classification variables, age group, city-type, and CNNM2 rs3740393 gene subgroup were all protective factors of magnesium deficiency. The OR value of Mg deficiency in 26–35 years and 36–45 years was 0.50 (*p* = 0.016) and 0.41 (*p* = 0.001) compared with that in 18–25 years. When compared with people in the city, the OR value of Mg deficiency in people of rural people was 0.57 (*p* = 0.045). In different genotypes of the CNNM2 rs3740393 gene, the OR values in comparison with carriers of both common alleles (G-G) were 0.56 for the haplotype G-C (*p* = 0.027) and 0.52 (*p* = 0.253) for the haplotype C-C.

### 3.4. Associations of Plasma Magnesium Concentration with Glucose Parameters

Table 4 presents the odds ratios for type 2 diabetes, glucose-hyperglycemia, and HbA1c-hyperglycemia associated with the levels of plasma magnesium concentrations as continuous variables and categorized into tertiles according to the distribution. Higher ORs for T2DM, glucose-hyperglycemia, and HbA1c-hyperglycemia were associated with lower plasma Mg concentrations. Magnesium less than 0.75 mmol/L is a risk factor for T2DM, glucose-hyperglycemia, and HbA1c-hyperglycemia (all the *p*-values were less than 0.05), the OR was 6.75, 4.02, and 7.68, respectively, and Adjustment for more potential confounders did not materially change the results. Meanwhile, the adjusted ORs for T2DM, glucose-hyperglycemia, and HbA1c-hyperglycemia across 1 mg/l higher plasma magnesium were 0.76 (95% CI, 0.65–0.90), 0.88 (95% CI, 0.77–0.99), and 0.72 (0.57–0.91), respectively.

## 4. Discussion

Magnesium is an essential element for normal vascular tone and insulin sensitivity [14]. Studies have shown that low magnesium status is associated with hypertension, coronary heart disease, type 2 diabetes, and metabolic syndrome [15]. Therefore, the assessment of magnesium status is consequently of great importance. At present, the indicators for evaluating magnesium nutritional status include magnesium or magnesium ion content in whole blood, serum/plasma, and urine; however, none of the biomarkers are considered perfect to evaluate the magnesium status. Serum or plasma magnesium concentration is still used as the standard one [3], although the normal range of plasma Mg varies from country to country, and different studies have used slightly different values. The normal plasma magnesium concentration is still defined as 0.76 to 1.15 mmol/L [16].

In this study, the median of plasma in Chinese childbearing women was 0.87 mmol/L. The prevalence of Mg deficiency and insufficiency was 4.69 and 33.09%, respectively. Compared with other countries, the magnesium deficiency rate in our study was much lower than that in Europe (50%) [17] and Germany (14.5%) [18]. However, compared with the study in Serbia [19], the magnesium deficiency rate in this study is higher than the 2.7% of Serbian adults, and the insufficiency rate is also higher than the 20.1% of Serbian adults. Although the above countries take 0.75 mmol/L (Germany 0.76 mmol/L) as the cut-off value of magnesium deficiency, the magnesium deficiency rate varies greatly between countries, ranging from 2.7 to 50%. This may be due to ethnic and economic differences between countries. Studies have shown that the deficiency rate of Mg is twice as common in women as in men [19,20], therefore, it is still necessary to pay more attention to women of childbearing age.

In the exploration of the risk factors of Mg deficiency in this study, the rural city-type was found to increase the risk of Mg deficiency, and it is also found that magnesium deficiency induces an increase in glucose. Compared to the people in the city area, people in the rural area were more prone to have Mg deficiency. This may be caused by different economic levels and eating habits between urban and rural areas. However, in the study of differences between rural and urban elements, there was no difference in the population distribution of Mg [21]. Magnesium deficiency also leads to an increase in glucose. For every unit of increase in blood glucose, the risk of magnesium deficiency increased by 1.61. This may be related to the regulatory effect of Mg on blood glucose metabolism. Magnesium can affect the tyrosine kinase activity of insulin receptors by transferring phosphate from ATP to protein so as to regulate the metabolism of glucose and insulin [22]. Therefore, it is of great significance to explore the relationship between magnesium and abnormal blood glucose. The results of our study still showed that Ca, LDL-C, age group of 26–35 years, 36–44 years, and CNNM2 rs3740393 genotypes of GC have been significant protective factors for childbearing women in China. Interestingly, contrary to previous studies which demonstrated that calcium competes with magnesium in functional transport of the crude ascending limb of the loop of Henle [23], calcium is found to be a protective factor for magnesium deficiency. This may be due to the low prevalence of magnesium deficiency in the study population. Differences in measurement responses between calcium and magnesium need to be verified in larger populations. The inconsistent result in comparison with a previous study [24] also occurred in the assessment of the relationship between LDL-c and Mg deficiency. As Adela Hruby’s study showed [25], the G allele of rs3740393 CNNM2 locus was associated with decreased serum magnesium, and GC genotype was a protective factor of magnesium deficiency in this study

In 1959, Beckett found for the primary time that there was a relationship between hypomagnesemia and type 2 diabetes [26]. This region has stirred extraordinary intrigued among researchers. Among the exploring relationships between Mg and glucose parameters analyzed herein, we confirmed significant evidence for an association with glucose parameters in <0.75 mmol/L group compared with ≥0.85 mmol/L group before and after adjusting for the confounding factors. When Mg was less than 0.75 mmol/L, the risk of T2DM (OR = 6.53, 95% CI 2.62–16.28), glucose-hyperglycemia (OR = 5.31, 95% CI 2.35–11.99), and HbA1c-hyperglycemia (OR = 9.60, 95% CI 2.96–31.09) was significantly increased in women of childbearing age. This result is consistent with a study of Korean adults, which found that serum magnesium levels were negatively correlated with fasting blood glucose levels [27]. In this study, regarding the dose–response analysis, we also found a significant interaction between plasma magnesium and glucose parameters. Each 1 mg/L increase in plasma magnesium was associated with a decreased risk of T2DM, glucose-hyperglycemia, and HbA1c-hyperglycemia. Our findings are generally consistent with results from some of the previous studies [14,28]. As for the mechanisms of how Mg regulates blood glucose metabolism, numerous experimental studies have been reported. In addition to the mechanisms mentioned above, magnesium can also influence phosphorylase b kinase activity by releasing glucose-1-phosphate from glycogen, or directly affecting glucose transporter activity 4 (GLUT4), which helps regulate glucose transport to cells [29,30].

Nutritional insufficiency is a zone of expanding significance within the field of medical science and has been found to be an imperative determinant of the broad scourge of chronic disease. This study is the first to evaluate and analyze the magnesium nutrition status of Chinese childbearing women based on nationally representative surveillance. We made a preliminary exploration of the deficiency rate of magnesium and its possible risk factors. In addition, in our study, we also discussed the relationship between magnesium nutritional status and blood glucose parameters and studied its detailed dose-response relationship. The limitation of this study lies in its cross-section study type, which could only observe the association but not the causal relationship between the variables. The narrow age range and population made it unclear whether this association would occur in other populations.

## 5. Conclusions

In summary, according to the latest monitoring results in 2015, the average level of magnesium in women of childbearing age in China is 0.87 mmol/L. Magnesium deficiency and insufficiency are common in this population, with a prevalence of nearly 40%. It is suggested that the risk of magnesium deficiency is higher in rural areas and people with hyperglycemia, and magnesium nutrition monitoring should be strengthened. In this study, plasma magnesium was independently and negatively associated with the incidence of T2DM, glucose-hyperglycemia, and HbA1c-hyperglycemia. Further studies are needed to confirm our findings and clarify the mechanisms behind this association.

## Figures and Tables

**Table 1 nutrients-14-00847-t001:** Plasma magnesium concentrations of 1895 childbearing women (mmol/L).

Characteristics		N	Median	P25	P75	F (t)	*p* Value
Total		1895	0.87	0.82	0.92		
Age group	18–25 years	656	0.87	0.82	0.92	2.367	0.094
	26–35 years	645	0.87	0.83	0.92		
	36–44 years	594	0.88	0.83	0.93		
District	Eastern	627	0.88	0.82	0.93	4.022	0.018
	Central	677	0.87	0.82	0.91		
	Western	591	0.88	0.83	0.92		
City-type	City	771	0.87	0.83	0.93	1.782	0.075
	Rural	1124	0.87	0.82	0.92		
Nationality	Han	1669	0.87	0.83	0.92	1.433	0.152
	Ethnic minorities	226	0.87	0.82	0.91		
BMI	Thin	145	0.88	0.84	0.93	1.925	0.123
	Normal	1062	0.88	0.83	0.92		
	Overweight	477	0.87	0.83	0.92		
	Obesity	211	0.86	0.81	0.92		
Education	Primary	479	0.88	0.83	0.92	2.774	0.063
	Medium	1092	0.87	0.82	0.92		
	Advanced	324	0.88	0.84	0.93		
Drink	Yes	1495	0.88	0.82	0.92	0.943	0.346
	No	400	0.87	0.83	0.91		
CNNM2 rs3740393	GG	984	0.87	0.82	0.91	5.883	0.003
	GC	777	0.88	0.83	0.92		
	CC	134	0.88	0.84	0.94		

**Table 2 nutrients-14-00847-t002:** Comparison of Mg status among different groups in 1895 subjects (%, 95% CI).

Characteristics		Deficiency (Mg < 0.75 mmol/L)		Insufficiency (0.85 mmol/L > Mg ≥ 0.75 mmol/L)		Sufficiency (Mg ≥ 0.85 mmol/L)		*p*-Value
		%	95% CI	%	95% CI	%	95% CI	
Total		4.69	3.74–5.65	33.09	30.97–35.21	62.22	60.03–64.40	<0.001
Age group	18–25 years	5.95	4.13–7.76	34.30	30.66–37.93	59.76	56–63.51	0.306
	26–35 years	4.19	2.64–5.73	32.40	28.79–36.02	63.41	59.69–67.13	
	36–44 years	3.87	2.32–5.43	32.49	28.72–36.26	63.64	59.76–67.51	
District	Eastern	3.99	2.45–5.52	32.22	28.56–35.88	63.80	60.03–67.56	0.118
	Central	5.76	4–7.52	35.75	32.13–39.36	58.49	54.78–62.21	
	Western	4.23	2.61–5.85	30.96	27.23–34.7	64.81	60.95–68.66	
City-type	City	3.37	2.1–4.65	32.43	29.12–35.73	64.20	46.93–51.44	0.054
	Rural	5.61	4.26–6.95	33.54	30.78–36.3	60.85	11.52–14.55	
Nationality	Han	4.73	3.71–5.75	33.07	30.81–35.33	62.19	59.86–64.52	0.979
	Ethnic minorities	4.42	1.74–7.11	33.19	27.04–39.33	62.39	56.07–68.71	
BMI	Thin	4.83	1.34–8.32	25.52	18.41–32.62	69.66	62.17–77.15	0.197
	Normal	4.52	3.27–5.77	32.20	29.39–35.02	63.28	60.38–66.18	
	Overweight	4.82	2.9–6.75	34.80	30.52–39.08	60.38	55.98–64.77	
	Obesity	5.21	2.21–8.22	38.86	32.28–45.45	55.92	49.22–62.63	
Education	Primary	4.59	2.72–6.47	31.94	27.76–36.12	63.47	59.15–67.78	0.251
	Medium	5.04	3.74–6.33	34.71	31.88–37.53	60.26	57.35–63.16	
	Advanced	3.70	1.65–5.76	29.32	24.36–34.28	66.98	61.85–72.1	
Drink	Yes	4.82	3.73–5.9	32.84	30.46–35.23	62.34	59.88–64.8	0.835
	No	4.25	2.27–6.23	34.00	29.35–38.65	61.75	56.98–66.52	
CNNM2 rs3740393	GG	5.69	4.24–7.14	34.65	31.68–37.63	59.65	56.59–62.72	0.063
	GC	3.73	2.4–5.07	31.92	28.64–35.2	64.35	60.98–67.72	
	CC	2.99	0.1–5.87	28.36	20.72–36	68.66	60.8–76.52	

**Table 3 nutrients-14-00847-t003:** Multivariate logistic regression model for risk factors associated with Mg deficiency.

Variables		β	OR	95% CI	*p*-Value
BMI (kg/m^2^)		0.03	1.04	(0.96–1.11)	0.334
CRP (mg/L)		−0.04	0.97	(0.87–1.08)	0.517
Ca (mmol/L)		−5.13	0.01	(0.00–0.02)	0.001
Blood Pressure (mmHg)	SBP	0.01	1.00	(0.98–1.03)	0.753
	DBP	0.01	1.01	(0.97–1.05)	0.568
Lipid (mmol/L)	TC	0.67	1.94	(0.78–4.83)	0.152
	Tg	0.05	1.05	(0.77–1.44)	0.748
	HDL-C	−0.01	0.99	(0.30–3.32)	0.990
	LDL-C	−1.06	0.35	(0.14–0.87)	0.024
Blood glucose	Glucose (mmol/L)	0.47	1.61	(1.31–1.97)	0.001
	HbA1c (%)	−0.20	0.82	(0.57–1.19)	0.295
Age group	18–25 years	1	1		
	26–35 years	−0.70	0.50	(0.28–0.88)	0.016
	36–44 years	−0.88	0.41	(0.22–0.78)	0.006
District	Eastern	1	1		
	Central	0.39	1.48	(0.84–2.63)	0.177
	Western	0.29	1.34	(0.70–2.54)	0.374
City-type	City	1	1		
	Rural	0.57	1.78	(1.01–3.11)	0.045
Drink	Yes	−0.12	0.89	(0.49–1.61)	0.703
	No	1	1		
Nationality	Han	1	1		
	Ethnic minorities	−0.11	0.90	(0.43–1.90)	0.782
Education	Primary	1	1		
	Medium	0.06	1.06	(0.59–1.91)	0.850
	Advanced	0.12	1.13	(0.47–2.70)	0.787
CNNM2 rs3740393	GG	1	1		
	GC	−0.57	0.56	(0.34–0.94)	0.027
	CC	−0.66	0.52	(0.17–1.60)	0.253

**Table 4 nutrients-14-00847-t004:** Associations of plasma magnesium concentration with glucose parameters.

Variables	Plasma Mg Concentrations (mmol/L)			*p* Trend	Per 1 mg/L of Mg	*p-*Value
	<0.75	0.75–0.85	≥0.85			
T2DM						
Model 1	6.75 (2.83–16.11)	1.68 (0.83–3.38)	Reference	0.001	0.74 (0.62–0.88)	0.001
Model 2^#^	6.53 (2.62–16.28)	1.65 (0.80–3.39)	Reference	0.001	0.76 (0.65–0.90)	0.002
Glucose-hyperglycemia						
Model 1	4.02 (1.98–8.13)	1.18 (0.71–1.97)	Reference	0.002	0.86 (0.76–0.98)	0.023
Model 2*	5.31 (2.35–11.99)	1.31 (0.75–2.28)	Reference	0.001	0.88 (0.77–0.99)	0.006
HbA1c-hyperglycemia						
Model 1	7.68 (2.77–21.27)	1.37 (0.56–3.43)	Reference	0.001	0.73 (0.58–0.91)	0.004
Model 2^&^	9.60 (2.96–31.09)	1.29 (0.48–3.55)	Reference	0.001	0.72 (0.57–0.91)	0.005

Model 2^#^: Adjusted for district, age group, Tg, CRP and Ca; Model 2*: Adjusted for district, age group, Tg, CRP, BMI, Ca and rs3740393; Model 2^&^: Adjusted for district, age group, Ca, SBP and CRP.

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
