# Peer review of "Magnesium Nutritional Status, Risk Factors, and the Associations with Glucose Parameters of Childbearing Women in the China Adult Chronic Disease and Nutrition Surveillance (2015)"

_nutrients, 2022, doi:10.3390/nu14040847_

Round 1

Reviewer 1 Report

Authors write:"The current cut-offs of plasma-Mg refer to [13]" :  this is a paper from 1986 and does not describe the cut-offs used  in the present study (lines 100-103) ! - As recently proposed by Castello & Elin et al." there is a need for an evidence-based reference interval for serum Mg (Adv Nutr 2016; 13: 977". Authors are asked to comment on this basic problem (see e.g.:  Orlava S et al.: BMC Pregnancy Childbirth 2021; 21: 76 pp. , or Bertingto et al. Nutrients 2017; 17: 296) .

I  fully agree with the cut-offs used - but the problem must be discussed to understand  differences reported in the literature! - 

Author Response

Response to reviewer 1

Dear professor,

We very appreciate your careful reading of our manuscript and the valuable suggestions. We have carefully considered the comments and revised the manuscript accordingly. The comments can be summarized as follows:

Point :The current cut-offs of plasma-Mg refer to [13]" :  this is a paper from 1986 and does not describe the cut-offs used  in the present study (lines 100-103) ! - As recently proposed by Castello & Elin et al." there is a need for an evidence-based reference interval for serum Mg (Adv Nutr 2016; 13: 977". Authors are asked to comment on this basic problem (see e.g.:  Orlava S et al.: BMC Pregnancy Childbirth 2021; 21: 76 pp. , or Bertingto et al. Nutrients 2017; 17: 296) .

I  fully agree with the cut-offs used - but the problem must be discussed to understand differences reported in the literature! - 

Response : Thank you so much for your kind suggestion. We have carefully read the literature you recommended and revised the wrong reference (reference 13). This really give us a lot of inspiration and help. As Apostoli[1] indicated the reference range for an element should be established periodically and separately because they can differ from gender to gender and be affected by dissimilar environmental scenarios. Aiming at this population, taking the health outcome as the outcome index, our group selected 182 healthy women and established the normal reference value range of magnesium for Chinese women of childbearing age[2]. In addition, we will explore the appropriate cut-off value of magnesium for disease prevention by combining more appropriate statistical methods with the prediabetes and type 2 diabetes as outcomes.

  • Apostoli P, Minoia C, Hamilton EI. Significance and utility of reference values in occupational medicine. Sci Total Environ. 1998 Jan 8;209(1):69-77.
  • Zhang, H.; Cao, Y.; Song, P.; Man, Q.; Mao, D.; Hu, Y.; Yang, L. Suggested Reference Ranges of Blood Mg and Ca Level in Childbearing Women of China: Analysis of China Adult Chronic Disease and Nutrition Surveillance (2015). Nutrients 2021, 13, 3287.

Reviewer 2 Report

This manuscript essentially just confirms that magnesium deficiency increases the risk for the metabolic syndrome and diabetes.  The novelty basically is that it has been found in another population group.  This manuscript needs improvement English syntax and word usage.  Other questions and comments that the authors should consider follow:

Abstract:  Line 21 – I do not understand how plasma glucose could be a risk factor for magnesium deficiency.  Most other studies state the reverse, that is, magnesium deficiency induces an increase in plasma magnesium.  This statement should be revised.

Introduction:  The second paragraph needs to be revised.  Line 37 – What is meant by “slightly” deficiency?   The signs of indicated for this deficiency suggests that it would be at least moderate to severe.  Line 38-39 – The signs of severe deficiency also could be indicated for chronic latent magnesium deficiency.  Line 41 – It should be noted that the prevalence of magnesium deficiency depends upon how it is defined.  Lines 43-44 – It would be better to state that magnesium deficiency contributes to the risk for diabetes.

Materials and Methods:  Line 76 – How was the serum obtained?  Line 84 - The method for HbA1c determination should be referenced.  Line 102 – Serum concentrations ≤0.75 mmol/L are not always asymptomatic.  Line 106 – Define GWAS.  Line 109 – Define SNP.

Results:  Lines 129, 137, - use the word “are” instead of “were”.  Table 2 – The total percentage does not add up to 100% in the Rural group, is the value 13.03 correct?  It is surprising that overweight and obesity did not give a significant effect – they look different than thin and normal.  Line 148 – See comments in abstract section.  Lines 150-151 – What is meant by protective factors?  Are you saying the magnesium requirement is less with these gene groups? 

Discussion:  First sentence makes no sense – please rephrase.  Line 194 – Could the higher percentage in Europe be the result of a different definition given for magnesium deficiency? Some thoughts should be given for such a large difference.   Line 200-201 and lines 206-207 – See comments in abstract above.   Lines 211-216 – Have you considered the possibility that magnesium deficiency results in lower calcium because it affects vitamin D utilization – not that calcium is protective against magnesium deficiency?

Author Response

Response to reviewer 2

Dear professor,

We very appreciate your careful reading of our manuscript and the valuable suggestions. We have carefully considered the comments and revised the manuscript accordingly. The comments can be summarized as follows:

Point 1: Abstract: Line 21 – I do not understand how plasma glucose could be a risk factor for magnesium deficiency. Most other studies state the reverse, that is, magnesium deficiency induces an increase in plasma magnesium. This statement should be revised.

Response 1: Thank you so much for the kind suggestion. We revised the statement in line 22-23.

Point 2: Introduction:  The second paragraph needs to be revised.  Line 37 – What is meant by “slightly” deficiency?   The signs of indicated for this deficiency suggests that it would be at least moderate to severe. Line 38-39 – The signs of severe deficiency also could be indicated for chronic latent magnesium deficiency. Line 41 – It should be noted that the prevalence of magnesium deficiency depends upon how it is defined. Lines 43-44 – It would be better to state that magnesium deficiency contributes to the risk for diabetes.

Response 2: Thank you very much for your suggestions. We carefully read the comments and revised the second paragraph in line 38-49.

Point 3:Materials and Methods:  Line 76 – How was the serum obtained?  Line 84 - The method for HbA1c determination should be referenced.  Line 102 – Serum concentrations ≤0.75 mmol/L are not always asymptomatic.  Line 106 – Define GWAS.  Line 109 – Define SNP.

Response 3: Thank you for your careful and kind suggestions. We have carefully revised the methodology in line 80-85, 90-91,109,113 and 116.

Point 4:Results:  Lines 129, 137, - use the word “are” instead of “were”.  Table 2 – The total percentage does not add up to 100% in the Rural group, is the value 13.03 correct?  It is surprising that overweight and obesity did not give a significant effect – they look different than thin and normal.  Line 148 – See comments in abstract section.  Lines 150-151 – What is meant by protective factors?  Are you saying the magnesium requirement is less with these gene groups? 

Response 4: Thank you so much for the suggestions. We have carefully revised the language problem in line 137, 145, 156-157. We are very sorry for the errors in the form, and have carefully checked and corrected the data in the form. In CNNM2 rs3740393 locus, people with GC and CC genotypes had a lower risk of magnesium deficiency than those with GG genotype. Therefore, GC and CC genotypes in  CNNM2 rs3740393 gene locus are protective factors of magnesium deficiency.

Point 5:Discussion:  First sentence makes no sense – please rephrase.  Line 194 – Could the higher percentage in Europe be the result of a different definition given for magnesium deficiency? Some thoughts should be given for such a large difference.   Line 200-201 and lines 206-207 – See comments in abstract above.   Lines 211-216 – Have you considered the possibility that magnesium deficiency results in lower calcium because it affects vitamin D utilization – not that calcium is protective against magnesium deficiency?

Response 5: Thank you so much for the suggestions. We have revised the problem of language expression in line 188,213 and 218.

For the difference of magnesium deficiency rate between different countries, we analyzed it in lines 206-209. Although the countries take 0.75mmol/L (Germany 0.76mmol/L) as the cut-off value of magnesium deficiency, the magnesium deficiency rate varies greatly between countries, ranging from 2.7% to 50%. This may be due to ethnic and economic differences between countries. 

For the relationship between calcium, magnesium and vitamin D, we also supplemented it in line 230-232. In this study, vitamin D has not been detected due to the limitation of blood sample volume, so the relationship between these three indicators can not be further analyzed. As for the relationship between calcium and magnesium deficiency, contrary to the research report, it may be because the magnesium deficiency rate in this study is relatively low, which can not fully reflect the relationship. Differences in measurement responses between calcium and magnesium need to be verified in larger populations.
